**communications** engineering

# A Metal Hydride Compressor Concept using Hydrogen as a Heat Transfer Fluid
Lukas Fleming [1,2], Maximilian Passing [1,2], Julián Puszkiel [1,2] ✉, Thomas Klassen[1,2] & Julian Jepsen [1,2]

Metal hydride hydrogen compressors have been explored as an alternative to mechanical hydrogen compressors since the first patents were filed in the 1970s. As heat engines, their productivity notably depends on the achievable heat transfer rate, which is limited by the pressure-bearing walls separating the heat transfer fluid from the reactive metal hydride beds and their effective thermal conductivity. Here we present and analyze an alternative metal hydride compressor system that uses hydrogen as a heat transfer fluid in direct convective contact with the metal hydride material. Following this principle, we demonstrate how an integrated compressor can be designed and how it behaves at both system and metal hydride bed levels. Simulations of a system operating at 10 – 90 °C indicate that specific productivities of 300 $L_n$ $h^{-1}$ $kg^{-1}$ can be achieved at low electrical energy demand, with isothermal efficiencies surpassing the ~75 % typically attained by mechanical piston compressors.

In the effort to reduce the emissions of greenhouse gases responsible for global climate change, hydrogen has been regarded as an essential energy vector[1–3]. Due to its low volumetric energy density in atmospheric conditions (0.003 kWh $L^{-1}$), the design of efficient hydrogen storage and transportation systems is still one of the most challenging bottlenecks in broad application[4]. Many options have been explored to improve the volumetric energy density of hydrogen: liquefaction, (cryo-)compressing, adsorption on carbon structures and metal-organic frameworks, chemical bonding in liquid organic hydrogen carriers, and metal hydrides[5]. Among these, the compressed hydrogen gas technology is the most mature and commercially available technology for hydrogen storage and transportation[6,7]. It can thus be expected that the compression of hydrogen will continue to play an essential role in the use of hydrogen as an energy vector in the near future.

Hydrogen compression from a gaseous source occurs using mechanical compressors in most applications. They have been used for decades for industrial purposes and are characterized by various types and sizes. In addition to many advantages, they are also associated with disadvantages such as electrical energy requirements (2–4 kWh $kg^{-1}$ for compression to 350 bar at refueling stations)[8], high maintenance requirements, high noise levels, and contamination of the hydrogen gas, which are problematic in some applications, such as hydrogen refueling stations[9]. Metal hydride compressors (MHC) have recently received increased attention to counter these problems[9–13]. As heat engines, they can be operated primarily with thermal energy, which, in the best case, is available as waste heat, for example, in metallurgical industries[14]. Due to the absence of moving parts,

they also have the potential to reduce maintenance costs while increasing reliability and operating with low noise emissions[15].

However, efficient heat transfer to the metal hydride (MH) bed is required to match the high power densities of mechanical compressors. In MH tanks of a size close to the application, the dynamic of the hydrogen uptake and release depends on the heat management[13]. One reason is the generally low effective thermal conductivity of the hydride-forming alloys, ranging from 0.1 to 1 W $m^{-1}$ $K^{-1}$, particularly when the material is in powder form[16]. The low effective thermal conductivity plays a very noticeable role in larger tanks with longer distances of heat conduction in the hydride bed[17]. Secondly, the necessary separation of the heat transfer fluid, usually a liquid or steam at temperatures above 100 °C, and the metal hydride bed must be overcome[18]. Regardless of whether the heating and cooling fluid flow is applied from outside of the pressure vessel or through an internal heat exchanger, heat must pass through the components, which are comparatively massive due to the high operating pressures. Another disadvantage is the inert thermal mass of these components, which contributes to the transient losses of typically around 30% of the exergy of the heat input, depending on the design[12,19].

The main challenge in conventional MH compressors is the sluggish heat transfer through the bed since it is primarily a heat conduction mechanism assisted by external forced convection (in indirect contact with a heating/cooling fluid). Improving heat transfer and reducing thermal masses is crucial for achieving high efficiencies in metal hydride compressors[6,9,11–13,15,18].

[1]Faculty of Mechanical and Civil Engineering, Helmut-Schmidt-University, Hamburg, Germany. [2]Institute of Hydrogen Technology, Helmholtz-Zentrum Hereon, Geesthacht, Germany. ✉e-mail: julian.puszkiel@hereon.de

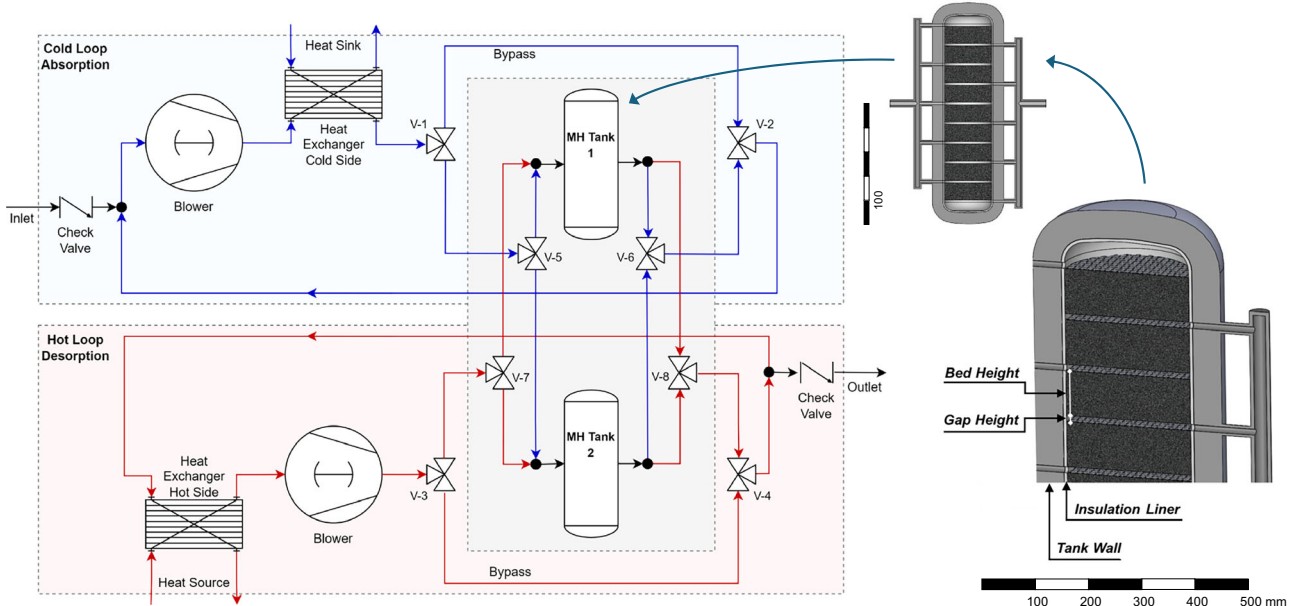

**Fig. 1 | Piping and instrumentation diagram of the Hydrogen Loop concept for a single-stage compressor (left) and cross-sectional view of one tank module (right).**

This paper presents a MH compressor concept that simultaneously uses hydrogen as a reactant and heat transfer medium. The concept was patented by Fleming et al., Fleming et al.[20,21]. So far, hydrogen as a heat transfer fluid has been studied experimentally[22] and simulatively[23] for use in MH storage tanks only. Existing patents[24,25] describe the operation of metal hydride storage systems using hydrogen as a heat transfer fluid. However, their system designs do not apply to hydrogen compression, as they do not enable simultaneous charging and discharging of multiple MH tanks or address the issue of large internal volumes (dead space). To the best of the authors' knowledge, this work represents the first analysis of a compressor system based on the dual utilization of hydrogen as a compressed product and heat transfer fluid, avoiding the additional design of heat exchangers integrated into the pressure vessel. Gaseous hydrogen, which has the highest specific heat capacity among all gases of 14.3 kJ kg$^{-1}$ K$^{-1}$, is cycled through a loop consisting of a blower, heat exchanger and the metal hydride bed for direct convective heat transfer. A compressor system operating according to this principle, referred to below as the Hydrogen Loop, was conceptualized (Fig. 1). It integrates two of the aforementioned loops that separately cycle cool hydrogen through one tank to drive the exothermic absorption reaction and hot hydrogen through the other tank to provide heat for the endothermic desorption reaction. Heat is taken out or introduced into the system during this process via gas-to-liquid heat exchangers. For every mol of hydrogen absorbed or desorbed, the static pressure in the respective loop drops or rises, causing hydrogen to replenish through the inlet or overflow through the outlet check valves. After the absorption and desorption reaction is completed, the gas phase pressure of the two tanks is first equilibrated, and then the connection of the two loops to the tanks is switched over by 3-way valves with T-bore V-5 to V-8. To avoid the equilibration of all pressure zones throughout the major void volumes of both loops, a bypass is created by valves V-1 to V-4 right before the switch-over and then closed again afterward.

The Hydrogen Loop concept is demonstrated and analyzed utilizing system simulations, for which a dynamic and pressure-driven model, including all of the components displayed in Fig. 1, was developed. The achievable productivity by hydrogen throughput is examined as a basis for comparison with traditional MH compressors using a liquid heat transfer fluid, hereafter referred to as conventional MHCs. In addition to comparing the productivity, the electrical energy required for the blower in the hydrogen loop is analyzed and put in relation to mechanical compressors.

## Results
### Model development for the hydrogen loop compressor
The productivity and efficiency of the proposed concept depend on the pressure loss to be overcome in the MH bed and the other components. For the heat transfer, it is also essential to represent the hydrogenation process dynamically. Finally, it was necessary to investigate the internal volumes in the individual loops. For these reasons, a fully pressure-driven and dynamic model library was developed in Aspen Custom Modeler®.

A sub-model was created for the MH tank because it contains the critical processes of hydrogen flow through the MH powder bed, the hydrogenation reaction, and heat transfer in porous media. The inside of the tank was designed as a series of disk-shaped MH beds of low height to reduce the pressure loss (Fig. 1). Due to the vertical flow of hydrogen through the beds and the insulation of the inner tank walls, a one-dimensional (1D) model in axial direction was developed to represent the tank behavior. The validity of this 1D model using the Finite Difference Method (FDM) was confirmed by comparison with 3D Finite Element Method (FEM) simulations in COMSOL Multiphysics® (Fig. 2). The reaction front (reacted fraction of the MH bed) moves through the bed in the shape of a plug flow, which favors using a 1D model. The rate of reaction and hydrogen flow are in good agreement, with R$^2$ values of 0.9976 and 0.9999, respectively. Close to the end of the reaction, the values for the 1D model are more drawn out than for the FEM model. Such deviation is due to the relatively low number of 10 nodes in the FDM chosen to reduce computational effort, while having a negligible impact on system performance. The convective heat transfer coefficient was calculated in both models and found to vary between 16 and 24 kW m$^{-2}$ K$^{-1}$ during the absorption and desorption reaction (Supplementary Fig. 7). For the pressure drop and heat transfer in the heat exchanger, design data for a plate heat exchanger are used[26]. The characteristics are very similar to a counterflow heat transfer and are defined accordingly in the model. The blower is modeled with an isentropic efficiency of 0.4 and a limitation of the maximum pressure difference to 1000 mbar, as is typical for Roots blowers. All pipes and valves include a pressure loss calculation. Heat exchange with the environment is not considered, as suitable conditions for thermal insulation of such a system are given. Thermal masses are only modeled for the MH material, as the tank wall is insulated from the inside, and most other components hardly experience any temperature fluctuations. The general layout can be described as pressure-driven and dynamic system modeling with embedded 1D models for the MH reactors.

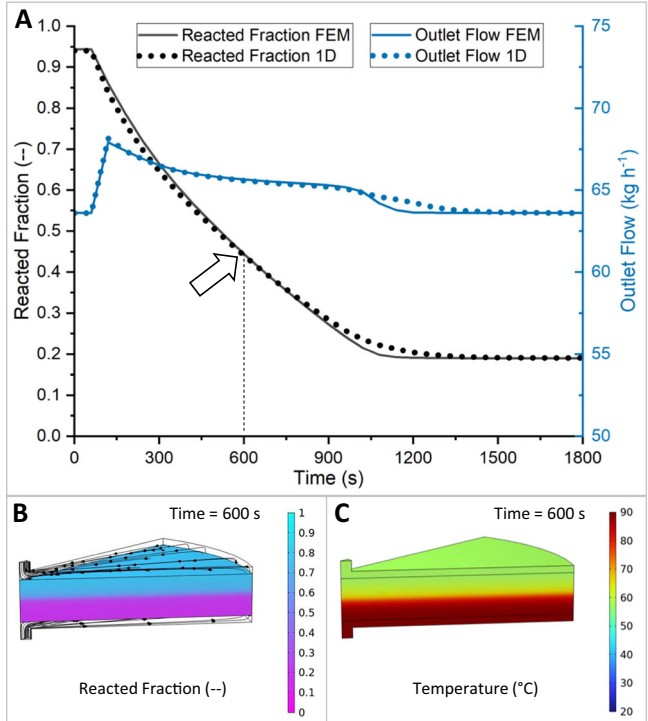

**Fig. 2 | Comparison of FEM and 1D model for the MH bed. A** Curves for reacted fraction and hydrogen flow rate for a desorption process. **B** FEM simulation yields purely vertical changes of reacted fraction and **C** temperature inside the flow-through MH bed.

coefficient of performance (CoP) is introduced in Eq. (1) that sets the ideal isothermal compression work $W_{isotherm}$ in relation to the electrical power $W_{el}$ consumed by the blower:

$$CoP = W_{isotherm}/W_{el} \qquad (1)$$

The heating and cooling power required to drive the dehydrogenation and hydrogenation reactions in both a conventional MHC and the proposed concept is assumed to be supplied by an external heat source and heat dissipation. These energy contributions are not included in the present evaluation. Consequently, the CoP can be greater than 1.

### Case studies

The productivity and CoP strongly depend on the thermodynamic properties of the MH materials and their alignment with the chosen operating pressure levels. Generally, the inlet and outlet pressures should be well within the equilibrium pressure range of the respective alloy at the absorption and desorption temperatures to ensure that a substantial fraction of the hydrogen capacity can be utilized in the process. Consequently, evaluating compressor performance for a single set of operating parameters can be misleading if the selected pressures do not accurately reflect the thermodynamic behavior of the materials. To provide a more comprehensive assessment, simulations were performed for a range of inlet and outlet pressures while maintaining constant temperature levels of the heating and cooling streams at 90 °C and 10 °C. Suitable conditions can be determined by placing distinct operating windows within the span of the pressure–composition isotherms (PCI), as illustrated in Fig. 3. Three such windows were defined by first selecting pairs of inlet and outlet pressures corresponding to compression ratios between two and three, labeled a)–c) for the low-pressure (LP) material and d)–f) for the high-pressure (HP) material, thereby forming the lower and upper boundaries. The window width was then determined by their intersection with the PCI curves and reduced by 5% on each side to avoid regions of slow reaction kinetics.

In the simulation setup for each case, a)–f), the pressure levels (window height) were controlled by fixed boundary conditions in the compressor inlet and outlet lines (compare Fig. 1). The usable capacity $\Delta c$ (window width) was maintained by triggering the switch-over between heating and cooling once both tanks reached the upper or lower limit of hydrogen saturation, depending on whether they were in the absorption or desorption phase, respectively. Table 1 summarizes the main parameters and material properties.

Three characteristic curves of CoP versus productivity were obtained for each material, as shown in Fig. 4. Both materials exhibit a non-linear behavior: the productivity increases with higher blower power, but the CoP decreases. The operating windows b) and e) show the highest level of productivity and efficiency, which can be explained by the high range of usable capacity and their central positioning between the absorption and desorption equilibrium pressure curves. In general, the curves of the HP setup show a better performance than the LP setup. As a benchmark for mechanical compressors, the ideal isothermal efficiency of 0.75 (made available by NEUMAN & ESSER)[31] is added to the diagram as a horizontal line K. Conversely, the compressor productivity should be compared to other compressors based on metal hydrides. The values of two experimental MHCs[32,33] are inserted as vertical lines L and M. It can be observed that the Hydrogen Loop concept is capable of operating at higher productivity than conventional MHCs. Although limited, there are also operating points that are not only more productive than conventional MHCs but also more electrically efficient than mechanical compressors.

### Sensitivity analysis

The productivity and CoP of the Hydrogen Loop compressor are influenced by a variety of parameters that can be categorized into:
- Thermodynamic (e.g., enthalpy of reaction) and kinetic (e.g., activation energy) properties of the MH material.

Two MH alloys were selected to demonstrate how the compressor concept works at certain pressure levels. For the lower pressure range, the commercially available alloy Hydralloy®C5[27] with thermodynamic data from[28] and a kinetic model from[29] is used. For the higher-pressure range, the alloy $Ti_{0.92}Zr_{0.10}Cr_{1.0}Mn_{0.6}Fe_{0.4}$ with thermodynamic and kinetic data from Zhan et al.[30] was selected. The total amount of MH material was set at 100 kg, with 50 kg per tank. Both alloys are $AB_2$–type alloys: intermetallic compounds known as Laves phase alloys, composed of a hydrogen-absorbing element in the A-site and a non-absorbing hydrogen transition element in the B-site, which acts as a catalyst. Such Laves phase alloys exhibit excellent long-term experimental stability, showing minimal degradation, which is mainly attributed to disproportionation. $AB_2$-type alloys with similar compositions to those proposed in this work exhibit only a 5% capacity loss after 2000 absorption/desorption cycles between 20 °C and 80 °C[28]. Therefore, the deviations due to material degradation upon absorption/desorption cycling can be neglected.

When simulating the Hydrogen Loop compressor, the time required to absorb the first tank and desorb the second and vice versa may differ due to varying temperature differences between the reaction equilibrium and the external heating and cooling sources. Furthermore, the hydrogen densities are different for the hot and cold loops, which affects the hydrogen mass flow and, hence, the heat transfer. These deviations can be compensated for by tuning the shaft power of each blower individually.

### Productivity and efficiency

As a measure of the productivity of the Hydrogen Loop, the output in standard liters per hour and per kilogram of MH material [$L_n$ $h^{-1}$ $kg^{-1}$] was chosen. While conventional MHCs, in principle, can operate without electrical energy consumption (excluding power requirements for actuators, sensors, and pumps for cycling the heat transfer fluid), the Hydrogen Loop concept presented here incorporates a blower that is considered a major electrical power consumer. To be able to compare the power consumption with that of a mechanical compressor, such as a piston compressor, a

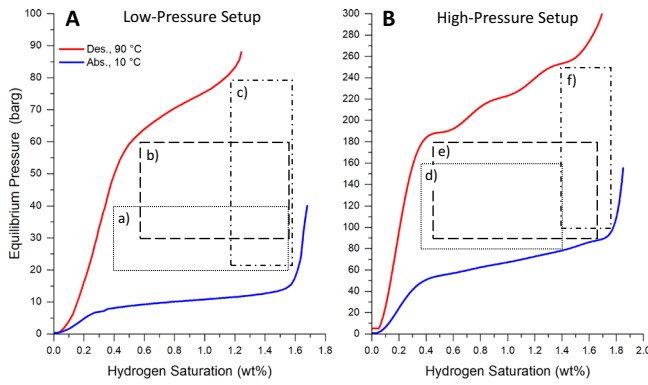

**Fig. 3 | Operational windows for different pressure levels.** The pressure–composition isotherms for **A** Hydralloy®C5 and **B** Ti0.92Zr0.10Cr1.0Mn0.6Fe0.4 of the absorption at 10 °C and desorption at 90 °C limit the reversible capacity range that can be obtained at certain inlet and outlet pressures. The windows are set off from their intersection with the equilibrium curves by a distance of 5% of their reversible capacity range to avoid slow kinetics at the end of every absorption and desorption process.

- Operational parameters, such as hydrogen pressure levels and the temperatures of the external heating and cooling sources.
- System inherent parameters that result from the engineering design, such as the pressure vessel diameter and height of the MH bed.
- Further physical properties of the MH material, such as porosity, particle size, and heat conductivity.

While in the first two categories, either the properties of an alloy are unalterably predetermined or the influence of temperatures, for example, appears clear, the effects of the system inherent parameters or certain assumptions for physical properties of the material are not so intuitively recognizable. For that reason, a sensitivity analysis using the Morris Screening Method[34] was carried out to investigate the effect of different parameters on the CoP of the system when the blower power was kept constant at a moderate level. A set $P_i$ of seven input parameters was chosen to represent the main design features and critical material assumptions:

- $P_1$: Particle diameter, uniform among all particles
- $P_2$: Solid thermal conductivity of the MH material
- $P_3$: Height of the MH porous bed
- $P_4$: Porosity of the MH bed
- $P_5$: Efficiency of the hydrogen blower

## Table 1 | Operational parameters and material properties

| Parameter | Units | | | |
|---|---|---|---|---|
| Low-Pressure Setup | | a) | b) | c) |
| Inlet pressure | barg | 20 | 30 | 23 |
| Outlet pressure | barg | 40 | 60 | 79 |
| Compression ratio | -- | 2 | 2 | 3.4 |
| Usable capacity Δc | wt% | 1.16 | 1.03 | 0.49 |
| Alloy composition | | Hydralloy®C5, Ti0.95Zr0.05Mn1.46V0.45Fe0.09 [27,28] | | |
| Enthalpy of absorption ΔH_abs | kJ mol⁻¹ | −20.8[28] | | |
| Enthalpy of desorption ΔH_des | kJ mol⁻¹ | 26.4[28] | | |
| Hydrogen capacity c_max | wt% | 1.73[28] | | |
| Crystalline density ρ_cryst | kg m⁻³ | 6395[28] | | |
| Porosity (unsaturated) ε₀ | -- | 0.6 | | |
| Particle diameter d_p,0 | μm | 20 | | |
| Thermal conductivity k_s | W m⁻¹ K⁻¹ | 10.1[28] | | |
| Specific heat capacity c_p | J kg⁻¹ K⁻¹ | 500 | | |
| Specific surface a_sf | m² g⁻¹ | 1.46 | | |
| High-Pressure Setup | | d) | e) | f) |
| Inlet pressure | barg | 80 | 90 | 100 |
| Outlet pressure | barg | 160 | 180 | 250 |
| Compression ratio | -- | 2 | 2 | 2.5 |
| Usable capacity Δc | wt% | 1.03 | 1.21 | 0.41 |
| Alloy composition | | Ti0.92Zr0.10Cr1.0Mn0.6Fe0.4 [30] | | |
| Enthalpy of absorption ΔH_abs | kJ mol⁻¹ | −15.71[30] | | |
| Enthalpy od desorption ΔH_des | kJ mol⁻¹ | 20.8[30] | | |
| Hydrogen capacity c_max | wt% | 1.85[30] | | |
| Crystalline density ρ_cryst | kg m⁻³ | 6647[30] | | |
| Porosity (unsaturated) ε₀ | -- | 0.6 | | |
| Particle diameter d_p,0 | μm | 20 | | |
| Thermal conductivity (solid) k_s | W m⁻¹ K⁻¹ | 12[30] | | |
| Specific heat capacity c_p | J kg⁻¹ K⁻¹ | 500 | | |
| Specific surface a_sf | m² g⁻¹ | 1.46 | | |

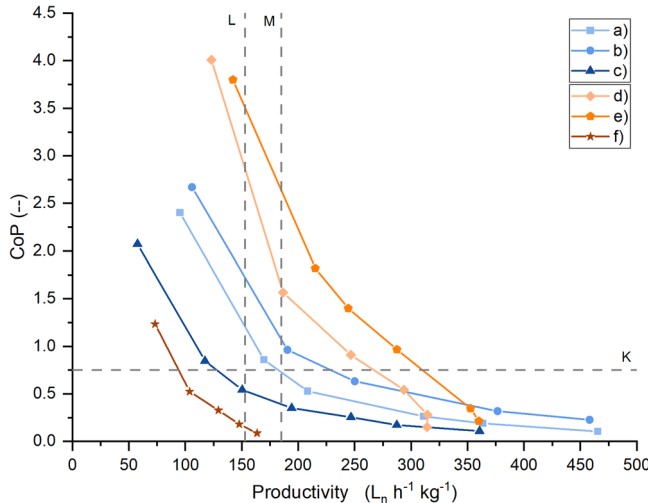

**Fig. 4 | Coefficient of performance (CoP) over productivity for the low–pressure and high–pressure setup.** Blower power was varied for three different operational windows (**a**–**c**) for the low-pressure setup and (**d**–**f**) for the high-pressure setup, resulting in 6 curves. Reference lines correspond to the isothermal efficiency of 0.75 (K)[31] for mechanical compressors and productivity values of experimental metal hydride compressors (L and M)[32,33].

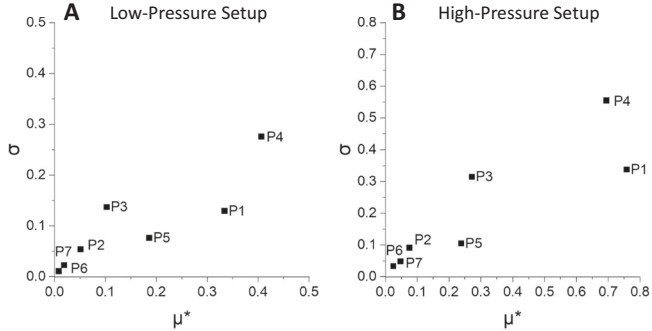

**Fig. 5 | Results of Morris Method for the low and high–pressure setup.** The elementary effects of parameters P1 to P7 for the low–pressure (left) and high–pressure (right) setup are plotted as σ over μ*.

- $P_6$: Void volume of the cold and hot loop
- $P_7$: Height of the gaps between beds inside the MH tanks

The results are displayed in a graph that relates the standard deviation of the elementary effects, σ, to the mean of the absolute value of the elementary effects, μ* (Fig. 5). Large values of $\mu_i^*$ indicate a high influence of the $i$th input parameter on the output. Small values of $\sigma_i$ suggest a linear relationship between the $i$th input and the output, while large values of $\sigma_i$ indicate nonlinear effects and/or interaction with one or more other inputs.

In both setups, the solid thermal conductivity, the void volumes of the loops and the additional dead space inside the tanks (gap height) have a negligible influence on the CoP of the system. Porosity and particle diameter have the most substantial effect, which can be attributed to the hydrogen flow resistance that determines the hydrogen flow through the bed and, thus, the heat transfer rate. The nonlinear and/or interactive characteristic of porosity (similar values of σ and μ*) can be explained by the fact that it factors quadratically and cubically into pressure drop calculation using the Ergun Eq. (10). Surprisingly, the blower efficiency has a comparatively small influence on the system performance. The equally moderate influence of the bed height can be

explained by its linear representation in the Ergun equation. Despite the low values of μ*, the bed height appears to interact with other factors, resulting in elevated values of σ.

## Discussion

The modeling and simulation work on the Hydrogen Loop compressor concept demonstrates that using hydrogen as a heat transfer fluid is suitable for increasing the productivity of MHCs to more than 200 $L_n$ $h^{-1}$ $kg^{-1}$ through enhanced heat transfer rates. Simultaneously, in the region of CoP > 0.75, it can achieve the same hydrogen compression work with less electrical energy compared to mechanical compressors. The productivity and CoP can be further increased with a higher quality of the available heating and cooling streams, directly affecting the amount of heat that can be transferred to or from the MH material. The convective heat transfer coefficient ranged between 16 and 24 kW $m^{-2}$ $K^{-1}$, demonstrating the potential for very high heat transfer rates when sufficient hydrogen flow is provided. This represents a considerable advantage compared to the conduction mechanism of a conventional MHC, which remains limited by the effective thermal conductivity of the material and the low effective cross-sectional area available for heat transfer. The sensitivity analysis additionally confirms the negligible impact of the solid heat conductivity of the MH material. Another mechanism to increase the CoP and productivity can be found in the hydrogen pressure level. Although the gas pressure and material were changed when comparing the LP and HP setups, it is reasonable to argue that high gas densities lead to higher mass flows, which favors heat transfer. The first term in the Ergun-Equation for laminar flow is relevant due to low hydrogen flow velocities of 0.005 to 0.02 m $s^{-1}$. The dynamic viscosity $\mu$ only slightly increases with pressure, whereas the density and the hydrogen mass flow increase according to real gas behavior. This results in higher convective heat transfer and a moderate increase in flow resistance.

The additional void volumes introduced by the blower, heat exchanger, and pipes to the system initially represented a concern for the productivity of the compressor. After improving the switch-over process to be isolated from the outer hot and cool loop, the outer volumes have a low impact on efficiency and productivity ( < 5% when varied in the value range for the sensitivity analysis), as can be observed in the sensitivity analysis and from the understanding of the flow diagram. When a conventional MHC is switched over from heating (desorption) to cooling (absorption), the hydrogen in the gas phase inside the tank gets absorbed first, which reduces the effective output (productivity). In the Hydrogen Loop compressor, however, when equalizing the two tanks during switch–over, this leads to charging and discharging the opposite tank above or under their final concentration value, respectively. When connecting the two, the high-pressure gas inside the just desorbed tank leads to overcharging the just absorbed tank. This effectively adds to the usable capacity that would have been available with the inlet and outlet pressure levels alone.

The Hydrogen Loop concept can be extended to include more than two parallel tanks to adjust the amount of MH material to the blower capacity. More sequential stages can be added by connecting the inlet of an HP stage to the outlet of an LP stage, where both stages are built, as shown in Fig. 1. This work focuses on demonstrating the compressor principle and analyzing a single stage. The coupling of multiple stages might lead to drawbacks in the CoP when the thermodynamic properties of both alloys lead to a reduction in usable capacity or when non-matching kinetics prolong the transition phase (as is the case for conventional MHCs, too).

From the sensitivity analysis, the porosity of the bed (flow resistance) was shown to have the highest influence on the CoP, while the dead space inside the tank, investigated by parameter $P_7$, has a low impact ( < 6%). This outcome opens up the prospect of an improved bed design with reduced flow resistance. Such freedom in design could also be used to counteract other phenomena, such as inhomogeneity resulting from the agglomeration of particles.

From an engineering perspective, it can be stated that all the main components, the heat exchangers, piping, valves and tanks, are either

available or manufacturable to the required specifications, including the static pressure-resistance. The hydrogen blower is the only exception because the pressure hull of the blowers available today must be designed for higher ratings.

## Methods

### 1D FDM Model in Aspen Custom Modeler®

The MH tanks are designed as flow-through reactors, with hydrogen moving through the fixed MH bed while transferring heat and reacting with it. This complex procedure is represented in a 1D sub-model for improved spatial resolution while remaining integrated into the system model. The discretization method applied is a $1^{st}$ order backward finite difference with a total of 10 nodes. The Implicit Euler method was used as the integrator, and the Mixed Newton method was used as the non-linear solver.

A schematic of the spatial node distribution within the bed is provided in Supplementary Fig. 3. While the MH tank contains multiple disc-shaped beds of height $L$ stacked vertically and connected to the hydrogen lines in parallel, the model simplifies this arrangement by representing it as a single bed. Owing to the demonstrated plug-flow behavior, the horizontal dimension can be chosen arbitrarily. Therefore, the model represents a single equivalent disc of the same height $L$, whose diameter $D$ is adjusted to yield the total mass of MH material according to its bulk density.

### Heat transfer in the porous metal hydride bed

A substantial flow rate is needed when hydrogen is used to transport the heat connected to the thermal mass and enthalpy of formation in a MH bed. Therefore, a non-thermal equilibrium is assumed when modeling the convective heat transfer between the hydrogen gas as fluid and the porous MH bed as a solid. At the same time, it is assumed that the convective heat transfer mechanism is much stronger than the conductive one.

The heat transfer model is governed by Eqs. (2) and (3), where $x$ is the direction of hydrogen flow, and the indices $s$ and $f$ denote the solid MH porous bed and the hydrogen as the fluid, respectively. Furthermore, $Q_s$ is the specific heat of the reaction and $u_f$ is the superficial velocity of the hydrogen flow.

$$(1 - \varepsilon)\rho_s c_{p,s}\frac{\partial T_s}{\partial t} = (1 - \varepsilon)k_s\frac{\partial^2 T_s}{\partial x^2} + q_{sf}\left(T_f - T_s\right) + (1 - \varepsilon)Q_s \quad (2)$$

$$\varepsilon\rho_f c_{p,f}\frac{\partial T_f}{\partial t} = \varepsilon\rho_f c_{p,f}u_f\frac{\partial T_f}{\partial x} + \varepsilon k_f\frac{\partial^2 T_f}{\partial x^2} + q_{sf}\left(T_s - T_f\right) \quad (3)$$

The interstitial convective heat transfer coefficient $q_{sf}$ was calculated for the bulk bed using Eqs. (4) to (6), where $a_{sf}$ is the specific surface area and $h_{sf}$ is the interstitial heat transfer coefficient.

$$q_{sf} = h_{sf}a_{sf} \quad (4)$$

$$\frac{1}{h_{sf}} = \frac{d_p}{\lambda_f Nu_{sf}} + \frac{d_p}{\beta\lambda_s} \quad (5)$$

$$Nu_{sf} = 2.0 + 1.1\,Pr^{1/3}Re \quad (6)$$

All variables associated with calculating the heat transfer coefficient were determined dynamically, but as average values for the entire MH bed. The coefficient $\beta$ was set to 10 for spherical particles. A BET analysis was performed for the specific surface area. For the cycled (40 °C and 40 bar of hydrogen) HydralloyC5® material and sieved in particle size between 20-30 microns, a value of $1.46\ m^2\ g^{-1}$ was measured. For the HP material, the source does not contain this information, which is why the same specific surface was used for comparison. The particle diameter $d_p$ and the bulk porosity $\varepsilon$ were calculated acc. to Eq. (7 − 9) as a function of the hydrogen saturation $\alpha$

and base values $d_{p,0}$ and $\varepsilon_0$, which are given as input parameters.

$$d_p = d_{p,0} * \left(1 - \alpha\,\Delta\rho\right)^{-1/3} \quad (7)$$

$$\varepsilon = \varepsilon_0 * \frac{1 - \frac{\alpha\,\Delta\rho}{\varepsilon_0}}{1 - \alpha\,\Delta\rho} \quad (8)$$

$$\Delta\rho = \frac{\rho_{cryst,\,unsat} - \rho_{cryst,\,sat}}{\rho_{cryst,\,unsat}} \quad (9)$$

### Pressure drop inside the porous MH bed

The Ergun Eq. (10), often used for fixed-bed reactors, was applied to calculate the pressure loss over the entire bulk bed. However, regarding the superficial velocities $u_f$ occurring in the modeled bed, only the first term (Kozeny-Carman) responsible for laminar flow becomes relevant. Spherical particles were assumed.

$$\Delta p_{bed} = \frac{180 * \mu * L * (1 - \varepsilon)^2}{d_p^2 * \varepsilon^3} * u_f \frac{1.75 * L * \rho_{H2} * (1 - \varepsilon)}{d_p * \varepsilon^3} * u_f * |u_f| \quad (10)$$

The total pressure drop $\Delta p_{bed}$ was then distributed equally over the height of the bed. The effect of this simplification, which is the local operative pressure for the hydrogenation reaction and the density, was considered negligible when compared to the overall static pressure and the relative density change of the gas.

### FEM model for verification

A single tank model for comparison was set up in Aspen Custom Modeler® version 12.1 and COMSOL Multiphysics version 6.2 with identical kinetic[29] and thermodynamic[35] model equations and geometry. The geometry was built as a 2D axisymmetric model. A mesh of triangles and rectangles was used at the wall interface to represent the fluid flow field. The skewness quality of the mesh showed a minimum value of 0.215 and an average quality of 0.847, having a total number of 38,940 elements and an element-to-volume ratio of 7.43E-4. The heat transport and fluid dynamic variables were solved using the PARDISO solver. The following three physical modules were used:

- Heat transport in porous media with local thermal inequality
- Free fluid flow and flow in porous media using the Brinkmann equation
- Transport of diluted species

### Sensitivity analysis using the Morris screening method

The R statistical computing environment and its package "sensitivity"[36] were used to create the sample matrix and calculate the results using the Morris Method. The number of factors is 7 ($P_1$–$P_7$) with 4 repetitions and scaling of the inputs activated. The selected parameter values for $P_1$–$P_7$, along with the complete sample matrix, are reported in the Supplementary Information.

### Further components and system model

Heat losses are not accounted for in the model. The tanks have an insulation layer on the inner pressure vessel wall. All other components can be fitted with external insulation and are exposed to a reasonably constant temperature level, eliminating the influence of inert heat capacities. Real gas behavior (Peng-Robinson) is used, and no gravitational forces are considered.

Gas compression with an isentropic efficiency of 0.4 is used as a base value for parameter $P_5$ to model the hydrogen blower. The low maximum pressure increase of Roots blowers for hydrogen is limited to 1000 mbar.

The electric motor attached to the blower is set to have a mechanical efficiency of 0.9.

A counterflow operating characteristic is assumed for the heat exchanger and an approximate dimensioning for a duty of up to 40 kW is specified. The pressure drop poses an uncertainty in the model since no continuous calculation model was found for the given application. Instead, manufacturer[26] calculated the heat exchanger at the system operating set points to derive a pressure loss coefficient. All pipes and valves contain a pressure loss calculation according to the Darcy-Weisbach equation or using a pressure loss coefficient.

The complete simulation setup in Aspen Custom Modeler® is shown in Supplementary Fig. 2.

## Brunauer–Emmett–Teller (BET) area measurement
The hydride-forming alloy Hydralloy®C5 was activated to full capacity and cycled 5 times. Then, the material was slowly oxidized in air atmosphere, followed by a degassing procedure. A sample was measured using the TriStar® II Plus surface area and porosity analyzer (Micromeritics Instrument Corp.).

## Data availability
All data generated in the scope of this work are included in the article and its Supplementary Information. Additional data are available on request from the corresponding author. Source data underlying the figures in this article are provided at https://doi.org/10.5281/zenodo.18261689.

## Code availability
The code generated for the simulations in Aspen Custom Modeler® and Comsol Multiphysics® is available on request from the corresponding author.

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

## Acknowledgements

This research work is in the frame of the project Digi-HyPro, funded by dtec.bw—Digitalization and Technology Research Center of the Bundeswehr, which the authors gratefully acknowledge. dtec.bw is funded by the European Union—NextGenerationEU.

## Author contributions

Conceptualization: L.F.; Methodology: L.F., J.P.; Analysis: L.F., J.P., M.P.; Writing—original draft: L.F.; Writing—review and editing: L.F., J.P., M.P., J.J.; Supervision: J.P., J.J., T.K.; Funding Acquisition: J.P., T.K., J.J.

## Funding

## Competing interests

The authors declare no competing interests.
