## [Transparent Peer Review file · Communications Engineering]

A Metal Hydride Compressor Concept using Hydrogen as a Heat Transfer Fluid

Corresponding Author: Dr Julián Puszkiet

Version 0:

Reviewer comments:

Reviewer #1

(Remarks to the Author)

The manuscript describes a numerical investigation of a metal hydride hydrogen compressor in which hydrogen is not only the compressed product but also as a heat transfer fluid to enhance the heat transfer rate during the absorption and the desorption of the hydrogen itself. The investigation is of interest for the general reader not only because of the scientific content but also of the technical application, as proved by the filed patents. However, the communication strategy seems, at least to me, weak because, on the one hand, the manuscript has an awkward structure in terms of sections as well as section sequence and, on the other, the information is provided to the reader as it becomes needed, making the text like a storytelling. Therefore, I feel suggesting strongly the following.

-In general, grouping the information in thematic sections, with clear section titles.

-Review the manuscript structure inserting a section on the case studies as well as a section on the modelling, with subsections for the different models and validations. The modelling section is of particular importance and should be carefully organized. Also, please, explain much better the case studies a) to f) that are not clear in the present text.

-Review the section sequence in a logical order, for example in Introduction, Concept, Models or Methods, Case studies, Results, Discussion. Please, note this is only an exemplary sequence, so feel free to adopt any other possible sequence.

Additionally, I would also suggest addressing the following minor observations.

-The Abstract shall include also some quantitative results and conclusions.

-Acronyms shall be avoided as much as possible, and particularly in figures, for instance "LP-Setup", and titles, for instance "BET".

-The use of the possessive form 's is wrongly applied also to objects. Expressions like hydrogen's, blower's, or system's shall be avoided.

-Consider naming the coefficient of performance as "performance evaluation criterion", which is quite common, but not universal, in the field of heat transfer equipment.

-Provide a short reference to R software, as the wording "The R package" may not be understood by the general reader.

Reviewer #3

(Remarks to the Author)

The manuscript presents a novel metal hydride hydrogen compressor concept, termed the Hydrogen Loop, which uses hydrogen itself as the heat transfer fluid in direct convective contact with the hydride bed. The study includes detailed system-level and metal hydride bed simulations, validated via 1D and FEM modeling in Aspen Custom Modeler® and

COMSOL®. The results demonstrate that this concept can increase productivity beyond conventional metal hydride compressors while maintaining low electrical energy demand relative to mechanical compressors. The work concludes with a sensitivity analysis highlighting key design parameters influencing performance.

Highlights:

- * The dual use of hydrogen as both working gas and heat transfer medium is an innovative approach that addresses a long-standing limitation in hydride compressors (poor heat transfer).
- * The integration of dynamic system modeling with 1D and FEM verification demonstrates strong methodological rigor.
- * The concept has practical engineering relevance, potentially simplifying system design by removing internal heat exchangers.
- * The performance comparison (CoP vs. productivity) with both mechanical and conventional MH compressors is well-motivated.
- * The manuscript is well-organized, logically flowing from motivation to modeling, results, and discussion.
- * Technical terminology is appropriate and consistent.
- * The writing is concise, with only minor grammatical issues.

There are some weaknesses:

- * The treatment of heat losses as negligible may overestimate CoP under realistic conditions.
- * No uncertainty quantification or error propagation analysis is presented.
- * The hydrogen blower model assumes constant isentropic efficiency and neglects potential leakage and thermal losses.
- * The hydride kinetics are represented by literature data without discussing potential deviations under cyclic operation.
- * It is not clear if the bed height L corresponds to the height of one disc or all 10 discs.
- * Some references to prior patents could include a short description of technical overlap to clarify novelty.
- * The pressure drop calculation for the heat exchangers may affect the simulation results. These values were not given

Version 1:

Reviewer comments:

Reviewer #3

(Remarks to the Author)

I thank the authors for the clear explanations and the changes to the manuscript. I recommend it for publication.

20 November 2025

SungHoon Hur
Editorial Board Member
Communications Engineering

Dear Editor,

We have received your email dated October 29, 2025, regarding our manuscript COMMS-ENG-25-0490-T, titled "A Metal Hydride Compressor Concept using Hydrogen as a Heat Transfer Fluid," written by Lukas Fleming, Maximilian Passing, Julián Puszkiewicz, Thomas Klassen, and Julian Jepsen. We are thankful to the reviewers for their comments and suggestions. We have revised our paper in response to the reviewers' recommendations, and all changes are highlighted in yellow. Here are the changes that have been made and the points that have been addressed:

Reviewer #1

The manuscript describes a numerical investigation of a metal hydride hydrogen compressor in which hydrogen is not only the compressed product but also as a heat transfer fluid to enhance the heat transfer rate during the absorption and the desorption of the hydrogen itself. The investigation is of interest for the general reader not only because of the scientific content but also of the technical application, as proved by the filed patents. However, the communication strategy seems, at least to me, weak because, on the one hand, the manuscript has an awkward structure in terms of sections as well as section sequence and, on the other, the information is provided to the reader as it becomes needed, making the text like a storytelling. Therefore, I feel suggesting strongly the following.

Authors: We thank the reviewer for the evaluation of our work.

Reviewer #1

- In general, grouping the information in thematic sections, with clear section titles.
- Review the manuscript structure inserting a section on the case studies as well as a section on the modelling, with subsections for the different models and validations. The modelling section is of particular importance and should be carefully organized. Also, please, explain much better the case studies a) to f) that are not clear in the present text.

Authors: We thank the reviewer for the observations and added a subsection "Case Studies." In this section, we provide a detailed explanation of how the operating parameters for cases a) to f) were determined. Please see the rewritten paragraph below, as well as highlighted in the manuscript.

Case Studies

The productivity and CoP strongly depend on the thermodynamic properties of the MH materials and their alignment with the chosen operating pressure levels. Generally, the inlet and outlet pressures should be well within the equilibrium pressure range of the respective alloy at the absorption and desorption temperatures to ensure that a significant fraction of the hydrogen capacity can be utilized in the process. Consequently, evaluating compressor performance for a single set of operating parameters can be misleading if the selected pressures do not accurately reflect the thermodynamic behavior of the materials. To provide a more comprehensive assessment, simulations were performed for a range of inlet and outlet pressures while maintaining

constant temperature levels of the heating and cooling streams at 90 °C and 10 °C. Suitable conditions can be determined by placing distinct operating windows within the span of the pressure-composition isotherms (PCI), as illustrated in Fig. 3. Three such windows were defined by first selecting pairs of inlet and outlet pressures corresponding to compression ratios between two and three, labeled a)-c) for the low-pressure (LP) material and d)-f) for the high-pressure (HP) material, thereby forming the lower and upper boundaries. The window width was then determined by their intersection with the PCI curves and reduced by 5 % on each side to avoid regions of slow reaction kinetics.

In the simulation setup for each case a)-f), the pressure levels (window height) were controlled by fixed boundary conditions in the compressor inlet and outlet lines (compare Fig. 1). The usable capacity Δc (window width) was maintained by triggering the switch-over between heating and cooling once both tanks reached the upper or lower limit of hydrogen saturation, depending on whether they were in the absorption or desorption phase, respectively. Table 1 summarizes the main parameters and material properties.

Three characteristic curves of CoP versus productivity were obtained for each material, as shown in Fig. 4. Both materials exhibit a non-linear behavior: [...]

Authors: We have retained the Modeling section unchanged, ensuring continued compliance with the journal's formatting requirements. In "Results - Model Development for the Hydrogen Loop Compressor," we describe the scope of the models developed and show the validation results. The detailed model description can then be found in the Methods section (Journal formatting requirement) and, in our opinion, is divided into appropriate subsections.

Reviewer #1

- Review the section sequence in a logical order, for example in Introduction, Concept, Models or Methods, Case studies, Results, Discussion. Please, note this is only an exemplary sequence, so feel free to adopt any other possible sequence.

Authors: Although we fully understand that the reviewer considers the sequence of sections and reviewer cites as examples to be appropriate for the paper, we must adhere to the journal's guidelines regarding paper structure.

Reviewer #1

Additionally, I would also suggest addressing the following minor observations.

- The Abstract shall include also some quantitative results and conclusions.

Authors: We thank the reviewer for this comment and modified the abstract accordingly:

Metal hydride hydrogen compressors have been explored as an alternative to mechanical hydrogen compressors since the first patents were filed in the 1970s. As heat engines, their productivity significantly depends on the achievable heat transfer rate, which is limited by the pressure-bearing walls separating the heat transfer fluid from the reactive metal hydride beds and their effective thermal conductivity. This work presents and analyzes an alternative metal hydride compressor system that uses hydrogen as a heat transfer fluid in direct convective contact with the metal hydride material. Following this principle, we demonstrate how an integrated compressor can be designed and how it behaves at both system and metal hydride bed levels. Simulations of a system operating at 10 – 90 °C indicate that specific productivities of 300 L_n h⁻¹ kg⁻¹ can be achieved at low electrical energy demand, with isothermal efficiencies surpassing the ~75 % typically attained by mechanical piston compressors.

Reviewer #1

- Acronyms shall be avoided as much as possible, and particularly in figures, for instance “LP-Setup”, and titles, for instance “BET”.

Authors: We followed the reviewer's suggestion and avoided using abbreviations. Please, see the text and figures in the manuscript.

Reviewer #1

- The use of the possessive form 's is wrongly applied also to objects. Expressions like hydrogen's, blower's, or system's shall be avoided.

Authors: We are thankful for the observation. We modified this language issue.

Reviewer #1

- Consider naming the coefficient of performance as "performance evaluation criterion", which is quite common, but not universal, in the field of heat transfer equipment.

Authors: We have carefully considered the reviewer's suggestion and concluded that the term 'CoP' is the closest to our application. We utilize a heat source (and cooling source) to generate work that exceeds the purely electrical energy input.

Reviewer #1

- Provide a short reference to R software, as the wording “The R package” may not be understood by the general reader.

Authors: We corrected this reference as follows

Sensitivity Analysis using the Morris Screening Method

The R statistical computing environment and its package "sensitivity"³⁶ were used to create the sample matrix and calculate the results using the Morris Method.

Reviewer #3

The manuscript presents a novel metal hydride hydrogen compressor concept, termed the Hydrogen Loop, which uses hydrogen itself as the heat transfer fluid in direct convective contact with the hydride bed. The study includes detailed system-level and metal hydride bed simulations, validated via 1D and FEM modeling in Aspen Custom Modeler® and COMSOL®. The results demonstrate that this concept can increase productivity beyond conventional metal hydride compressors while maintaining low electrical energy demand relative to mechanical compressors. The work concludes with a sensitivity analysis highlighting key design parameters influencing performance.

Highlights:

- * The dual use of hydrogen as both working gas and heat transfer medium is an innovative approach that addresses a long-standing limitation in hydride compressors (poor heat transfer).
- * The integration of dynamic system modeling with 1D and FEM verification demonstrates strong methodological rigor.
- * The concept has practical engineering relevance, potentially simplifying system design by removing internal heat exchangers.
- * The performance comparison (CoP vs. productivity) with both mechanical and conventional MH compressors is well-motivated.
- * The manuscript is well-organized, logically flowing from motivation to modeling, results, and discussion.
- * Technical terminology is appropriate and consistent.
- * The writing is concise, with only minor grammatical issues.

Authors: We thank the reviewer for the positive assessment of our work.

Reviewer #3

There are some weaknesses:

- * The treatment of heat losses as negligible may overestimate CoP under realistic conditions.

Authors: We thank the reviewer for this observation. We agree that heat losses could affect the CoP in a way that the hot hydrogen loses temperature on its way to the tank and inside the tank, and part of the heat may be lost through the inner insulation of the walls. Ultimately, this would reduce the temperature driving force and slow down the desorption reaction rate.

However, the present study primarily aims to demonstrate the fundamental feasibility of the concept. For this purpose, an adiabatic system (except for the external heating and cooling sources) was applied as one of several modeling assumptions and boundary conditions.

We mentioned the section "Methods: Further Components and System Model", and as you can see below:

Further Components and System Model

Heat losses are not accounted for in the model. The tanks have an insulation layer on the inner pressure vessel wall. All other components can be fitted with external insulation and exposed to a reasonably constant temperature level, eliminating the influence of inert heat capacities.

Reviewer #3

* No uncertainty quantification or error propagation analysis is presented.

Authors: As this study is based on numerical simulations rather than experimental measurements, no direct measurement uncertainties or experimental error propagation can be quantified. However, two approaches were applied. First, the validity of the 1D model, developed using the Finite Difference Method (FDM), was verified by comparison with 3D Finite Element Method (FEM) simulations in COMSOL Multiphysics. The rate of reaction and hydrogen flow obtained from the 1D FDM model and 3D FEM model are in good agreement, with R^2 values of 0.9976 and 0.9999, respectively. Second, the weight of the most relevant parameters was evaluated through sensitivity analysis, which showed that the results of the model, characterized by the CoP, are primarily influenced by the porosity and diameter of the metal hydride bed particles. Such parameters are in good agreement with those reported in the literature (ref. [28] in the paper). Therefore, the sensitivity analysis enabled us to demonstrate the reduced degree of uncertainty resulting from the model parameter assumptions and to illustrate the robustness of the modeling approach. The first approach can be read in the results section and the second in the discussion section, as it is possible to read below:

Results: Model Development for the Hydrogen Loop Compressor

The **accuracy** of this 1D model using the Finite Difference Method (FDM) was confirmed by comparison with 3D Finite Element Method (FEM) simulations in COMSOL Multiphysics® (Fig. 2). The reaction front (reacted fraction of the MH bed) moves through the bed in the shape of a plug flow, which favors using a 1D model. The rate of reaction and hydrogen flow are in good agreement, with R^2 values of 0.9976 and 0.9999, respectively.

Discussion

From the sensitivity analysis, the porosity of the bed (flow resistance) was shown to have the highest influence on the CoP, while the dead space inside the tank, investigated by parameter P7, has a low impact (<6%).

Reviewer #3

* The hydrogen blower model assumes constant isentropic efficiency and neglects potential leakage and thermal losses.

Authors: We thank the reviewer for this well-considered comment. If leakage refers to internal backflow or blow-by of hydrogen through the small gap between the rotor and housing, this effect is implicitly accounted for in the model. It is reflected in the assumed isentropic efficiency and in the limitation of the maximum pressure difference to 1 bar, which corresponds to typical operating limits for Roots-type hydrogen blowers, where higher pressure differences promote backflow.

Regarding the modeling/implementation in Aspen Custom Modeler itself, the blower efficiency, as a technical specification, is represented as a fixed parameter during the simulation run. But as soon as the pressure difference inside the blower approaches the limit of 1 bar, the efficiency is lowered by an exponential curve. During a simulation run, this results in a blower output that corresponds to the pressure difference limit of 1 bar: whenever the pressure difference rises close to the limit, the output flow breaks down due to the reduced efficiency, which then again lowers the pressure drop in the system.

The influence of the blower isentropic efficiency level was systematically examined in the sensitivity analysis, where it was varied between 30% and 60%, showing only a moderate effect on the overall CoP. Thermal losses are difficult to quantify separately; they are implicitly included in the efficiency assumptions and discussed in connection with the previous comment on heat losses.

Furthermore, a mechanical efficiency of 0.9 was applied to represent the electric motor and drivetrain.

For clarification, the relevant passage from the manuscript is reproduced below.

Further Components and System Model

Gas compression with an isentropic efficiency of 0.4 is used as a base value for parameter P5 to model the hydrogen blower. The low maximum pressure increase of Roots blowers for hydrogen is limited to 1000 mbar. The electric motor attached to the blower is set to have a mechanical efficiency of 0.9.

Reviewer #3

* The hydride kinetics are represented by literature data without discussing potential deviations under cyclic operation.

Authors: We thank the reviewer for the valuable observation. The cycling stability of the material is a relevant property for applications such as thermal compression. Indeed, AB₂-type alloys are chosen for metal hydride compression applications due to proper thermodynamic properties, fast kinetic behavior, and excellent long-term experimental stability upon hydrogenation/dehydrogenation cycles. We added the following paragraph to the results section, clearly stating that further discussion about deviations due to stability issues is not within the scope of this work.

Results - Model Development for the Hydrogen Loop Compressor

Both alloys are AB₂-type alloys: intermetallic compounds known as Laves phase alloys, composed of a hydrogen-absorbing element in the A-site and a non-absorbing hydrogen transition element in the B-site, which acts as a catalyst. Such Laves phase alloys exhibit excellent long-term experimental stability, showing minimal degradation, which is mainly attributed to disproportionation. AB₂-type alloys with similar compositions to those proposed in this work exhibit only a 5% capacity loss after 2000 absorption/desorption cycles between 20 °C and 80 °C [31]. Therefore, the deviations due to material degradation upon absorption/desorption cycling can be neglected.

Reviewer #3

* It is not clear if the bed height L corresponds to the height of one disc or all 10 discs.

Authors: We thank the reviewer for this helpful comment. First, we would like to clarify our terminology and the definition of the bed height parameter L.

We see that the reviewer well understood that the tank contains multiple disks connected in parallel to the hydrogen inlet and outlet streams. This arrangement was made to maintain a reasonably low bed height for the entire MH material while achieving a realistic tank design. So the bed height L corresponds to the height of each disc. For the modeling, however, we simplified the arrangement by creating one large disc with height L that contains all the material. FEM simulations showed that a plug flow behavior is present, supporting this approach to modeling the bed. For the 1D model in ACM, the height of the bed was discretized into 10 nodes, each representing a (again disc-shaped) volume inside the MH bed. Please refer to the modified Figure 2 in the Supplementary Information (also shown below). There, the number of nodes was reduced to 6 for the sake of clarity in the Figure.

Then, secondly, we changed the corresponding paragraph in the Methods section:

Methods - 1D FDM Model in Aspen Custom Modeler®

While the MH tank contains multiple disc-shaped beds of height L stacked vertically and connected to the hydrogen lines in parallel, the model simplifies this arrangement. Owing to the demonstrated plug-flow behavior, the horizontal dimension can be chosen arbitrarily. Therefore, the model represents a single equivalent disc of the same height L, whose diameter D is adjusted to yield the total mass of MH material according to its bulk density.

Authors: Lastly, we would like to point out that Figure 1 in the manuscript indicates the bed height in the crosssectional view.

This is the changed Figure 3 in the Supplementary Information for more clarity:

Reviewer #3

* Some references to prior patents could include a short description of technical overlap to clarify novelty.

Authors: We thank the reviewer for the suggestion. We reviewed the corresponding sentence and revised it for greater clarity and a more detailed description of technical overlap; see below and the highlighted section in the manuscript.

6th paragraph, page 1:

Existing patents^{24,25} describe the operation of metal hydride storage systems using hydrogen as a heat transfer fluid. However, their system designs do not apply to hydrogen compression, as they do not enable simultaneous charging and discharging of multiple MH tanks or address the issue of large internal volumes (dead space).

Reviewer #3

* The pressure drop calculation for the heat exchangers may affect the simulation results. These values were not given.

Authors: To obtain realistic pressure-drop values for the hydrogen loop, we contacted manufacturers of gas-to-liquid plate heat exchangers and requested custom calculations for the hydrogen mass flow, change in temperature, and pressure levels corresponding to our simulation setup. A modeling approach was required because the dynamic simulation must compute the pressure drop for *any* occurring hydrogen mass flow, while retaining a realistic representation of the heat exchanger's internal gas volume.

Table 1 presents the derivation of a channel-wise pressure-loss coefficient ζ_{channel} based on manufacturer design calculations for the stainless-steel plate heat-exchanger series AXP27-AN (Alfa Laval; calculation sheets shown in Fig. 1). In these heat exchangers, the flow capacity can be adjusted by varying the number of plates, with each added plate contributing one gas-side or liquid-side channel in alternating order. The coefficient ζ_{channel} was determined with reference to one hydrogen channel, allowing the system to be scaled by varying the number of plates (channels) while preserving consistent hydraulic behavior. The calculation is based on Eq. (1), where the pressure drop Δp is given by the manufacturer's design calculation and the reference velocity is determined using a reference cross-sectional area derived from manufacturer specifications (green fields) and complementary geometric assumptions (blue fields).

Four custom calculations were obtained. Cases 1 and 2 originate from an early stage of the work and were prepared for a hydrogen mass flow of 5 kg h^{-1} with a wide assumed temperature swing ($\approx 10\text{--}90 \text{ }^\circ\text{C}$). Cases 3 and 4 were requested at a later stage, when the simulation setup was fully developed in Aspen Custom Modeler and a larger bed mass of 50 kg metal hydride per tank had been selected. By that time, we had also gained the insight that the hydrogen temperature swing, i.e. the difference between tank inlet and outlet, would be narrower during operation. Despite the differences in hydrogen mass flow and temperature change, all four design calculations yield a similar pressure loss coefficient in the range of approximately 190 to 240. This consistency underscores the robustness of the derived value.

In the second part of Table 1, ζ_{channel} is used to compute the pressure drop for a reference hydrogen mass flow of 100 kg/h , while adjusting the number of gas-side channels such that the internal gas volume of the heat exchanger equals 2 L . This

analysis shows that for every 100 kg/h of hydrogen flow only 2 L of heat-exchanger gas volume must be added, and the resulting pressure drop remains very low ($\approx 9\text{--}16$ mbar).

A heat exchanger of 2 L volume was implemented in the simulation. The total gas volume of one hydrogen loop, excluding the MH tanks, is ~ 14 L, consisting of the blower (10 L), the heat exchanger (2 L), and piping (~ 2 L). This value is used as the baseline for parameter P_6 (V_{outside}) in the sensitivity analysis. The sample-matrix range (± 80 % of the baseline) spans 3-25 L and therefore covers a wide spectrum of realistic engineering assumptions for the internal volumes of components such as the heat exchanger. The sensitivity analysis shows that these external volumes have a negligible effect on compressor performance, supporting the conclusion that heat exchangers in the Hydrogen Loop can be designed to minimize pressure drop without concern for their internal gas volume.

Upon reviewing the design calculations from Alfa Laval, we noticed that we had included the wrong reference [26] and corrected it.

Table 1: Determination of the channel-wise pressure-loss coefficient for the Alfa Laval AXP27AN plate heat exchanger

		Alfa Laval - Case 1 AXP27 AN-8H	Alfa Laval - Case 2 AXP27 AN-8H	Alfa Laval - Case 3 AXP27 AN-70H	Alfa Laval - Case 4 AXP27 AN-74H
		H2 heating from 20°C to 90°C at 5 kg/h -> desorption	H2 cooling from 90°C to 11°C at 5 kg/h -> absorption	H2 heating from 55°C to 90 °C at 125 kg/h -> desorption	H2 cooling from 50°C to 12 °C at 100 kg/h -> absorption
Deriving zeta-value per channel					
Number of channels	--	4,00	3,00	34,00	36,00
Plate height	mm	2,75	2,75	2,75	2,75
Channel length	mm	250,00	250,00	250,00	250,00
Channel width	mm	70,00	70,00	70,00	70,00
Manifold volume per channel	mm ³	19477,87	19477,87	19477,87	19477,87
Volume per channel	mm ³	50000,00	50000,00	50000,00	50000,00
Channel area	mm ²	17500,00	17500,00	17500,00	17500,00
Channel height	mm	1,74	1,74	1,74	1,74
Reference cross-sect. area per channel	mm²	122,09	122,09	122,09	122,09
Mass flow	kg/h	5,00	5,00	124,60	100,00
Hydrogen pressure	barg	30,00	30,00	61,00	30,00
Hydrogen density	kg/m ³	2,25	2,30	4,35	2,40
Volume flow	m ³ /h	2,22	2,17	28,64	41,67
Velocity per channel	m/s	1,26	1,65	1,92	2,63
Hydrogen pressure drop	mbar	4,00	6,00	19,40	19,20
Press. loss. coeff. ζ_{channel}	--	222,54	191,94	242,77	230,73
Scaling to 100 kg/h and 2 L volume					
Number of channels	--	40,00	40,00	40,00	40,00
Mass flow	kg/h	100,00	100,00	100,00	100,00
Hydrogen pressure	barg	30,00	30,00	61,00	30,00
Hydrogen density	kg/m ³	2,25	2,30	4,35	2,40
Volume flow	m ³ /h	44,44	43,48	22,99	41,67
Velocity per channel	m/s	2,53	2,47	1,31	2,37
Pressure drop	mbar	16,00	13,50	9,03	15,55
Total volume	liter	2,00	2,00	2,00	2,00

Technische Spezifikation

AlfaNova Wärmeübertrager

Projekt Referenz: Projekt 2
 Position: Case 2
 Typ: AXP27 AN-8H
 Artikel-Nr.: 3075150429
 Anz. Komponenten: 1

Seite: 1(2)

Prozessdaten	Warme Seite S4 -> S3		Kalte Seite S2 -> S1	
	Wärmeleistung:	kW 1,4		
Medium:	Water		Hydrogen	
Aufgaben Typ:	Liquid cooling		Gas heating	
Massenstrom:	kg/h 57,4		5,00	
Eintrittstemperatur:	°C 91,0		20,0	
Saugseitiger Druck:	bara 30,00		30,00	
Austrittstemperatur:	°C 70,0		90,0	
Saugseitiger Druck:	bara 30,00		30,00	
Gesamt Druckverlust berechnet (zulässig)	kPa 0,2 (30,00)		0,4 (30,00)	
Geschwindigkeit Anschlüsse:	m/s 0,04		1,43	
Marge kalkuliert (vorgegeben):			18(0) %	
Eff. Foulingwiderstand * 10000:	m² K/W 1,9		1,4	
LMTD:	K 11,6		12,1	
k-Wert, verschmutzte Beding.:	W/(m²·K) 805,9		873,4	

Apparatespezifikation	
Übertragungsfläche:	m² 0,2
Strömungsrichtung der Medien:	Countercurrent
Anzahl Platten:	8
Anzahl der Wege:	1 1 1
Kanalaufteilung:	1'3H 1'4H
Kanalvolumen:	dm³ 0,2 0,2
Anz. Kreisläufe:	1 1
Auslegungsdruck bei -10 °C:	bar 110 70
Auslegungsdruck bei 150 °C:	bar 110 70
Auslegungstemperatur min/max:	°C -10 / 150
Druckgeräterichtlinie:	PED
Material Kanalplatten/Dichtung:	ALLOY 316 / SS
Anschluss S4 (Warm-Ein):	Welding 33,7 mm ALLOY 316
Anschluss S3 (Warm-Aus):	Welding 33,7 mm ALLOY 316
Anschluss S2 (Kalt-Ein):	Welding 33,7 mm ALLOY 316
Anschluss S1 (Kalt-Aus):	Welding 33,7 mm ALLOY 316
Länge x Breite x Höhe:	mm 79 x 160 x 362
Leer-/Betriebsgewicht:	kg 0 x 0 / 21,81
Länge x Breite x Höhe (verpackt):	mm 0 x 0 x 0
Versandgewicht:	kg 21,7

Technische Spezifikation

AlfaNova Wärmeübertrager

Projekt Referenz Auslegung
 Typ AXP27 AN-70H
 Anz. Komponenten 1
 Artikelnummer 3075180070

Seite 1 of 1

Prozessdaten	Warme Seite S1 -> S2		Kalte Seite S3 -> S4	
	Wärmeleistung:	kW 17,0		
Medium:	Water		Hydrogen	
Aufgaben Typ:	Liquid cooling		Gas heating	
Massenstrom:	kg/h 1.481		124,6	
Eintrittstemperatur:	°C 90,0		55,0	
Saugseitiger Druck:	bara 30,00		61,00	
Austrittstemperatur:	°C 80,0		89,0	
Gesamt Druckverlust berechnet (zulässig)	kPa 1,19 (2,00)		1,94 (2,00)	
Geschwindigkeit Anschlüsse:	m/s 0,75		13,70	
Marge kalkuliert (vorgegeben):			36 (0) %	
Eff. Foulingwiderstand * 10000:	m² K/W 2,0		7,4	
LMTD:	K 7,4		10,9	
k-Wert, verschmutzte Beding.:	W/(m²·K) 1.350,8		766,5	

Apparatespezifikation	
Übertragungsfläche:	m² 1,7
Strömungsrichtung der Medien:	Countercurrent
Anzahl Platten:	70
Anzahl der Wege:	1 1 1
Kanalaufteilung:	1'35H 1'34H
Kanalvolumen:	dm³ 1,75 1,70
Anzahl der Kreise:	1 1
Auslegungsdruck bei (-10 °C):	bar 70 110
Auslegungsdruck bei (150 °C):	bar 70 110
Auslegungstemperatur min/max:	°C -10 / 150
Druckgeräterichtlinie:	PED
Material Kanalplatten/Dichtung:	ALLOY 316 / SS
Anschluss S1 (Warm-Ein):	Soldering 1 1/8" ALLOY 316
Anschluss S2 (Warm-Aus):	Soldering 1 1/8" ALLOY 316
Anschluss S3 (Kalt-Ein):	Soldering 1 1/8" ALLOY 316
Anschluss S4 (Kalt-Aus):	Soldering 1 1/8" ALLOY 316
Unit dimension (length x width x height):	mm 229 x 160 x 362
Leer- / Betriebsgewicht:	kg 30,73 / 32,42
Versandgewicht:	kg 30,73

Fig. 1: Alfa Laval custom calculations for a series AXP27AN heat exchanger

Technische Spezifikation

AlfaNova Wärmeübertrager

Projekt Referenz: Projekt 2
 Position: Case 2
 Typ: AXP27 AN-8H
 Artikel-Nr.: 3075150429
 Anz. Komponenten: 1

Seite: 1(2)

Prozessdaten	Warme Seite S4 -> S3		Kalte Seite S2 -> S1	
	Wärmeleistung:	kW 1,6		
Medium:	Hydrogen		Water	
Aufgaben Typ:	Gas cooling		Liquid heating	
Massenstrom:	kg/h 5,00		67,8	
Eintrittstemperatur:	°C 90,0		10,0	
Saugseitiger Druck:	bara 30,00		30,00	
Austrittstemperatur:	°C 11,0		30,0	
Gesamt Druckverlust berechnet (zulässig)	kPa 0,6 (30,00)		0,2 (30,00)	
Geschwindigkeit Anschlüsse:	m/s 1,77		0,05	
Marge kalkuliert (vorgegeben):			14(0) %	
Eff. Foulingwiderstand * 10000:	m² K/W 1,9		1,4	
LMTD:	K 11,6		12,1	
k-Wert, verschmutzte Beding.:	W/(m²·K) 805,9		873,4	

Apparatespezifikation	
Übertragungsfläche:	m² 0,2
Strömungsrichtung der Medien:	Countercurrent
Anzahl Platten:	8
Anzahl der Wege:	1 1 1
Kanalaufteilung:	1'3H 1'4H
Kanalvolumen:	dm³ 0,2 0,2
Anz. Kreisläufe:	1 1
Auslegungsdruck bei -10 °C:	bar 110 70
Auslegungsdruck bei 150 °C:	bar 110 70
Auslegungstemperatur min/max:	°C -10 / 150
Druckgeräterichtlinie:	PED
Material Kanalplatten/Dichtung:	ALLOY 316 / SS
Anschluss S4 (Warm-Ein):	Welding 33,7 mm ALLOY 316
Anschluss S3 (Warm-Aus):	Welding 33,7 mm ALLOY 316
Anschluss S2 (Kalt-Ein):	Welding 33,7 mm ALLOY 316
Anschluss S1 (Kalt-Aus):	Welding 33,7 mm ALLOY 316
Länge x Breite x Höhe:	mm 79 x 160 x 362
Leer-/Betriebsgewicht:	kg 0 x 0 / 21,81
Länge x Breite x Höhe (verpackt):	mm 0 x 0 x 0
Versandgewicht:	kg 21,7

Technische Spezifikation

AlfaNova Wärmeübertrager

Projekt Referenz Auslegung
 Typ AXP27 AN-74H
 Anz. Komponenten 1
 Artikelnummer 3075180065

Seite 1 of 1

Prozessdaten	Warme Seite S4 -> S3		Kalte Seite S2 -> S1	
	Wärmeleistung:	kW 15,0		
Medium:	Hydrogen		Water	
Aufgaben Typ:	Gas cooling		Liquid heating	
Massenstrom:	kg/h 100,0		1.287	
Eintrittstemperatur:	°C 50,0		10,0	
Saugseitiger Druck:	bara 30,00		61,00	
Austrittstemperatur:	°C 12,4		20,0	
Gesamt Druckverlust berechnet (zulässig)	kPa 1,92 (2,00)		0,905 (30,0)	
Geschwindigkeit Anschlüsse:	m/s 22,02		0,64	
Marge kalkuliert (vorgegeben):			80 (0) %	
Eff. Foulingwiderstand * 10000:	m² K/W 5,8		10,9	
LMTD:	K 10,9		10,9	
k-Wert, verschmutzte Beding.:	W/(m²·K) 766,5		766,5	

Apparatespezifikation	
Übertragungsfläche:	m² 1,8
Strömungsrichtung der Medien:	Countercurrent
Anzahl Platten:	74
Anzahl der Wege:	1 1 1
Kanalaufteilung:	1'36H 1'37H
Kanalvolumen:	dm³ 1,80 1,85
Anzahl der Kreise:	1 1
Auslegungsdruck bei (-10 °C):	bar 110 70
Auslegungsdruck bei (150 °C):	bar 110 70
Auslegungstemperatur min/max:	°C -10 / 150
Druckgeräterichtlinie:	PED
Material Kanalplatten/Dichtung:	ALLOY 316 / SS
Anschluss S4 (Warm-Ein):	Soldering 1 1/8" ALLOY 316
Anschluss S3 (Warm-Aus):	Soldering 1 1/8" ALLOY 316
Anschluss S2 (Kalt-Ein):	Soldering 1 1/8" ALLOY 316
Anschluss S1 (Kalt-Aus):	Soldering 1 1/8" ALLOY 316
Unit dimension (length x width x height):	mm 239 x 160 x 362
Leer- / Betriebsgewicht:	kg 31,25 / 33,10
Versandgewicht:	kg 31,25

Hoping the manuscript meets all editorial requirements, I remain,

Sincerely yours,

Dr. J.A. Puszkiel

Deputy Head of Department
 Group leader of Stationary Applications

Institute of Hydrogen Technology

Helmholtz-Zentrum Hereon

Geb. 33, Büro 206

T. +49 (0) 176 72599729

julian.puszkiew@hereon.de

Head of the laboratory

Applied Materials Engineering

Helmut-Schmidt-University

University of the Federal Armed Forces Hamburg

Geb. H1, Büro 1221

T. +49 (0) 40 6541 3297

m. +49 (0) 152 24127094

puszkiew@hsu-hh.de